# IconicITA: Iconicity ratings of the Italian affective lexicon

**Andrea Gregor de Varda**[1]☉*, **Tommaso Lamarra**[2]☉*, **Andrea Amelio Ravelli**[2], **Chiara Saponaro**[1], **Beatrice Giustolisi**[1], **Marianna Bolognesi**[2]

**1** Department of Psychology, University of Milano-Bicocca, Milan, Italy, **2** Department of Modern Languages, Literatures and Cultures, University of Bologna, Bologna, Italy

☉ These authors contributed equally to this work.
* a.devarda@campus.unimib.it (AGDV); tommaso.lamarra2@unibo.it (TL)

## Abstract

Iconicity, defined as the potential of linguistic signs to resemble properties or features of their referents, is increasingly recognized as a general property of language. One common approach for quantifying iconicity is to collect iconicity ratings. Although iconicity datasets have been developed for several languages, no comprehensive dataset of iconicity ratings is currently available for Italian. The current study presents *IconicITA*, the first dataset of Italian iconicity ratings for the 1,121 words of the Italian adaptation of Affective Norms for English Words (ANEW). Ratings were collected from both Italian native speakers (L1) and English native speakers with Italian as a second language (L2). Including L2 participants allowed us to contribute to the debate on whether iconicity ratings genuinely measure form-meaning resemblance, rather than exclusively reflecting semantic properties. We showed that L1 Italian iconicity ratings are positively associated with perceptual strength in the auditory and haptic modalities, and with specificity ratings. Conversely, we found a negative correlation between iconicity and concreteness, age of acquisition, word frequency, and letter frequency. In general, the relationship between Italian iconicity norms and various psycholinguistic variables largely replicated previous findings in the literature on iconicity. Considering L2 data, the ratings provided by L2 speakers correlated more strongly with the Italian L1 data compared to the translation-equivalent English L1 data. This finding suggests that participants' judgments were influenced not only by the semantic information of the words but also by language-specific form-level properties. We take this result as evidence of the validity of iconicity ratings to operationalize the degree of resemblance between words' form and meaning.

## Introduction

The arbitrariness of linguistic signs – that is, the idea that there is no inherent connection between a sign's form and meaning – has been a fundamental tenet in the

**Data availability statement:** Code and data are available online on OSF: https://osf.io/r2tqs/.

**Funding:** M.B., T.L., A.A.R. are Funded by the European Union (GRANT AGREEMENT: ERC-2021-STG-101039777). Views and opinions expressed are however those of the author(s) only and do not necessarily reflect those of the European Union or the European Research Council Executive Agency. Neither the European Union nor the granting authority can be held responsible for them. The funders had no role in study design, data collection and analysis, decision to publish, or preparation of the manuscript.

**Competing interests:** The authors have declared that no competing interests exist.

study of language. It is only in recent years that there has been a shift towards the recognition of *iconicity*, namely the potential of a linguistic sign to reproduce, resemble, and imitate qualities or features of the referent, as a crucial and general property of language [1–3]. Currently, linguists, psychologists, and cognitive scientists agree in considering the lexicon as constituted of elements whose form-meaning resemblance ranges from clearly motivated to completely arbitrary [1,4]. Within non-arbitrariness, at least two key concepts can be distinguished: iconicity, which serves as an umbrella term encompassing various linguistic phenomena such as onomatopoeias, ideophones, and gestures, and systematicity, defined as a "statistical relationship between the patterns of sound for a group of words and their usage" [1, p. 606].

Systematicity is an orthogonal notion to iconicity [5,6] and should not be conflated with it. For example, phonesthemes – regular form-meaning mappings [3] – illustrate systematicity but not iconicity. A representative case is the sequence *gl-*, which is commonly associated with shininess [7], as seen in words like *glitter* or *glimmer*. However, *gl-* is not iconic because there is no inherent, natural reason why this particular sequence should evoke a resemblance to the concept of "shininess." Conversely, the sequence *cl-* available in words like *click* or *clank* is both systematic and iconic, because in these words the form-meaning mapping relies on the perceptually grounded similarity to the metallic sound that the sequence *cl-* denotes.

Notably, it has been clarified that arbitrariness, systematicity, and iconicity all contribute to language development and communication [1]. Since arbitrary words lack a direct link to their perceptual referents through form-meaning mapping, they tend to create more general representations rather than specific instances, facilitating the learning of a large vocabulary [8] and enhancing the efficiency and discriminability of language [9]. Systematicity instead supports sound-category distinctions and produces positive effects in language processing by making speakers able to recognize the proper grammatical categories (verbs, nouns, adjectives) in the right context through systematic and regular prosodic and phonological cues [1]. Finally, iconicity further facilitates word learning [1,2,5,10,11] by establishing a more constrained relation between form and meaning, conveying sensorial information in the word's representation, and assisting the displacement – namely, the ability to refer to not immediately available objects [9].

Considering the advantages of iconicity to language learning and communication, it is not surprising that a renewed interest in this topic [5] has led researchers to investigate this general property using different empirical approaches [12–14], considering languages across modalities (e.g., [15,16]) and with cross-linguistic perspectives [16–20]. Significant developments within this research program have relied on human ratings to quantify the degree of iconicity of a given word [21]. Iconicity ratings have been collected for various languages (i.e., English [6,19,22]; Japanese [17]; Spanish [23]; ASL [24,25], BSL [20,26]) but, to the best of our knowledge, a dataset gathered in a systematic way for Italian is not currently available. Furthermore, no iconicity ratings for a spoken language have been collected from both native and L2 speakers in a systematic way, with the purpose of comparing the two populations and assessing the impact of language proficiency.

Considering spoken languages, iconicity ratings are usually collected by proficient users of a language (e.g., [6,23]), whereas ratings for sign languages come not only from proficient users, but also from raters with no knowledge of the target sign language (e.g., 16, 24]). This second possibility has recently been exploited in the analysis of spoken languages' iconicity as well [13,14]. Studies focusing on sign languages reveal that, even if ratings of signers vs. non-signers display a significant positive correlation, there are differences in the distribution of the ratings across the lexicon [27,28], in accordance with previous findings suggesting that the cultural and linguistic background of the raters has an impact on the ratings themselves (e.g., [28,29]).

To contribute to this line of discussion, the current study presents *IconicITA*, the first dataset of Italian iconicity ratings for the 1,121 words of the Italian adaptation [30] of Affective Norms for English Words (ANEW) [31]. We collected ratings from both Italian native speakers (L1) and English native speakers using Italian as a second language (L2). Furthermore, we leveraged second-language learners' judgments to evaluate the construct validity of iconicity ratings. By comparing ratings from Italian L2 speakers with those from native speakers of both Italian and English, we can assess whether iconicity ratings measure genuine form-meaning resemblance rather than purely semantic properties.

## Theoretical background

The first empirical studies drawing attention to iconicity in the cognitive sciences date back to the early 20th century [32–34]. In particular, the seminal studies by Köhler [32,33] and Sapir [34] were among the first to provide evidence of a cross-modal correspondence between form and meaning. These studies showed that participants consistently associate simple shapes (an angular and a rounded shape) with two non-words (*bouba-kiki* [35] or *takete-maluma* [32,33]). Critically, the angular shape was consistently linked to the non-words *takete* or *kiki*, whereas the rounded, blob-like shape was linked to the non-words *bouba* or *maluma*, suggesting that certain linguistic sounds may inherently convey specific sensory qualities related to the visual appearance of their referents. These results have been replicated cross-linguistically [36]. This kind of sound-shape mapping could simultaneously involve two or three sensory modalities – that is, visual, auditory, and haptic [36]. Furthermore, Sapir [34] found that words with back vowels are considered more appropriate to signal large objects, while words with front vowels tend to be considered pertinent to symbolize smaller objects. Notably, it has been proposed that the universality of these phenomena could depend on an imitative nature at the origin of *sound symbolism*, specifically the emulation of an external entity's shape by the shape of the mouth in the act of producing a specific sound [3]. In addition, meaningful form-meaning mappings could be observed in both prosodic elements (e.g., *loooong* to express an extension of duration [9,37]) and morphological processes, as for the linguistic phenomenon of reduplication (e.g., *gorogoro* in Japanese denotes a heavy object rolling repeatedly [10]). Finally, iconicity can reflect the "sequential order" of an event, as for the well-known Latin expression *veni*, *vidi*, *vici*, which iconically represents the order followed by the three actions [3].

One of the most critical aspects of iconicity is the benefit provided by iconic associations to word learning [2,9–11,18,19,38,39]. By depicting traits of the referent denoted, iconic words are easier to learn: the form-meaning resemblance in iconic signs or words, based on an analogical strategy [1,5,40,41] and sensory-based information [4], bootstraps the referential process that associates a word with its referent [9], and consequently, word learning. Indeed, empirical evidence demonstrates that iconic words are highly common in the earliest acquired lexicon, as evidenced by a negative correlation between age of acquisition and iconicity ratings [19,21,22]. They are more frequently produced in conversation by both children and adults speaking to children [11,38], and they are easier to recall compared to non-iconic words [42]. Moreover, iconicity may also have a positive effect on language processing, as iconic words were processed more quickly in both visual and phonological Lexical Decision Tasks ("Is this letter string a word?") in a recent study [43]. However, considering the same set of words, no significant effect of iconicity on the reaction times from the English Lexicon Project (ELP) [44] was observed [43]. The authors attributed this result to a different distribution of iconic words within the two datasets, since the proportion of iconic words in the authors' dataset was higher than in the ELP. Studies with sign

 

languages, where iconicity is pervasive, also present mixed results as far as the role of iconicity in word processing (e.g., [15,45,46]). Recent research suggests that iconicity does not boost lexical access per se but might play a role in decision making processes [45].

To make iconicity empirically observable and quantifiable, as for the studies discussed above, it is necessary to obtain reliable measures to operationalize the resemblance between form and meaning embodied in iconic words or signs. One possibility consists of gathering iconicity ratings by asking participants to rate on a Likert scale to what extent the sound of a word resembles its meaning [12,40]. Iconicity is not an all-or-nothing property; rather, it varies along a continuum. The use of Likert scales for collecting iconicity ratings naturally captures this aspect, reflecting graded differences in how strongly words resemble their meanings [29,40]. However, whether iconicity ratings truly measure the speakers' intuition of a form-meaning correspondence has been a debated subject. In fact, other semantic information could explain participants' performance in iconicity rating tasks [17,21]. Specifically, speakers may rate words based on their perceptual strength and the sensory information they convey rather than on a genuine comprehension of the form-meaning mapping. However, although statistically relevant, the correlation between sensory strength [47] and iconicity ratings is mild ($r = 0.18$), explaining only 3% of the variance in the latter [48]. This supports the idea that iconicity ratings are not entirely explained by perceptual strength, opening the door to the possibility that they may reflect other factors — possibly form-meaning resemblance ( [21]; see also [49]). Additionally, iconicity ratings correlate with objective measurements of similarity between word and natural sounds (e.g., the sound of the word "frog" and the noise produced by a frog), indicating that, when providing such judgments, participants sensitize to both the words' and the referents' sound properties [49]. Moreover, iconicity ratings have been proved valuable for identifying patterns within the lexicon and across different languages, offering useful insights for psycholinguistics research [21].

Previous research has found systematic relationships between iconicity ratings and various psycholinguistic variables. For instance, iconicity ratings display a positive relationship with sensory experience ratings, particularly in the auditory and tactile modalities, and only weakly in the visual, olfactory, and gustatory modalities [22]. Because of this relation with perceptual experiences, it has been thought that iconic words would be generally concrete [50]. However, evidence suggests the opposite: higher iconicity tends to be associated with lower concreteness scores [6], based on existing norms [51]. Iconicity also negatively correlates with semantic neighborhood density [4], word frequency [6,11,19], and letter frequency [6]. Finally, as mentioned above, iconic words tend to be acquired early, as reflected in a negative correlation with age of acquisition [19,21,22]. These findings motivate our investigation into whether similar patterns emerge for Italian iconicity ratings.

## Aims of the study

In the present study, we collected and analyzed perceived iconicity judgments for the 1,121 Italian words of the Italian adaptation of the ANEW dataset [30]. To evaluate whether iconicity ratings truly reflect the participants' appreciation of form-meaning relationships or whether they merely depend on semantic information alone, we collected iconicity judgments from two distinct groups: Italian native speakers (L1) and English native speakers learning Italian as a second language (L2). We formulated two sets of research questions, one linked to L1 ratings (RQs 1), and one to L2 ratings (RQs 2):

1) What is the relationship between perceived iconicity in Italian and other psycholinguistic variables and decision latencies? How does this compare to the results that have been reported for English?

Overall, we expect to replicate in Italian (L1) the patterns that have been reported in English, namely:

a) A positive relationship with sensory (and in particular, auditory and haptic) experience ratings.

b) A negative relationship with concreteness ratings.

c) A negative relationship with frequency.

d) A negative relationship with letter frequency.

e) A negative relationship with semantic neighborhood density.

f) A negative relationship with age of acquisition (AoA) ratings.

Additionally, we decided to investigate the relation between iconicity and specificity, i.e., a relational property reflecting how narrowly a conceptual category refers to its members [52–54]. Human-generated specificity judgments for the Italian language demonstrate that highly specific words are generally less frequent and tend to occur in a more limited variety of contexts that are overall semantically close to one another [55]. Given that iconic signs or words focus on depicting specific aspects of the referent rather than broader or more general properties, we hypothesize that more iconic words will typically encode more specific meanings and activate precise semantic features.

Regarding the influence of iconicity on chronometric data, we can hypothesize two possible scenarios. On the one hand, we might expect to find a facilitatory effect of iconicity (rated by L1 speakers) on lexical decision latencies for Italian [56], in line with the experiments reported in the aforementioned study [43]. On the other hand, iconicity did not exhibit a significant effect on reaction times [43] reported in the ELP dataset [44] for the same set of words used in those experiments. As reported by the authors, a list context might have influenced the difference in results. Similarly, the ANEW dataset includes a wide variety of words with plausibly different iconicity levels; therefore, if previous results were driven by list effects, we might also expect no significant effect of iconicity on decision latencies.

2) What's the relation between iconicity ratings provided in L1 and L2? Can this relationship inform the debate on whether iconicity ratings do really measure iconicity? How does language proficiency in L2 impact the relation between iconicity ratings in L1 and L2?

In general, we expect to find a positive correlation between iconicity ratings provided by L1 (Italian, English) and L2 (Italian) speakers. Moreover, we expect the strength of these correlations to be informative with respect to the construct validity of iconicity ratings. As previously discussed, it has been suggested that participants might rely on the meaning of a word alone when judging its degree of iconicity instead of focusing on form-meaning similarity [21]. If this is the case, correlations between Italian L2 ratings and both Italian and English L1 ratings should be comparable, as we are focusing on the same set of words, translated from English to Italian [30]. However, if iconicity judgments genuinely reflect sensitivity to language-specific form-level properties, correlations should be stronger between L1 and L2 Italian ratings than between L2 Italian and L1 English ratings. Consistent with this hypothesis, the correlation between L1 and L2 Italian ratings is expected to increase with proficiency: the more experience an L2 speaker has with Italian phonology and lexical meanings, the more their ratings should converge with those of native Italian speakers. Lastly, consistent with the proposal that language experience plays a pivotal role in the appreciation of iconic patterns in language [29], we hypothesize that greater proficiency in Italian among L2 speakers will entail an increased sensitivity to language-specific form-meaning resemblance, and ultimately greater alignment with L1 iconicity ratings.

## Methods

The present study was positively evaluated (Prot. N. RM-2023–628) by the local commission for minimal-risk studies of the Department of Psychology of the University of Milan-Bicocca, which operates under the mandate of the University Ethics Committee, and carried out according to the principles of the Declaration of Helsinki.

### Materials

The 1,121 words from the ANEW database were divided into 5 lists of 224 words per list (one list had 225 words). Moreover, we selected 30 additional words: 5 onomatopoeic words (like the word *bang*) included in all lists and used as control

stimuli and 25 phonosymbolic proper words (5 per list), which were supposed to receive higher iconicity ratings compared to the mean ratings of the ANEW words.

The 5 onomatopoeic words and the 25 phonosymbolic words were selected from the Tullio De Mauro's online dictionary (https://dizionario.internazionale.it/). At first, we selected the words classified as basic lexicon, that is, the set of words marked as "fondamentale"/FO (*fundamental*), "alto uso"/AU (*high use*), and "alta disponibilità"/AD (*high availability*). From this first set, we excluded all those words for which the etymological origin was not attested as onomatopoeic or was signaled as exoticism and loan word (loan translation, calque), obtaining a list of 504 Italian words characterized by a high degree of form-meaning resemblance. At this stage, we needed to distinguish onomatopoeic words from phonosymbolic ones, thus we asked two annotators (both Italian native speakers) to evaluate them. Focusing on words the annotators agreed on, we split the list into two groups and randomly selected 25 phonosymbolic words (5 for each list) and 5 pure onomatopoeic words to be inserted in the experiment.

## Participants

A total of 128 participants were recruited for this study, 59 L1 Italian native speakers and 69 L2 Italian learners, completing 160 annotation sessions (74 L1 and 86 L2). For the L2 group, we restricted the participation only to learners with English as their native language, in order to control for potential influences due to the participants' first language and to avoid differences in the perception of word iconicity based on their linguistic background. All participants were neurotypical, without any language disorder. Written informed consent was obtained from all participants who voluntarily agreed to participate in the task. All the L1 speakers and the majority of the L2 were recruited through the Prolific platform, and the remaining L2 participants were recruited via word-of-mouth among students and personal contacts. Participant recruitment began on December 7, 2023, and ended on April 10, 2024.

To pre-select L2 speakers and to evaluate their proficiency in Italian, we ran a prescreening study. Participants who took part in the prescreening had to declare that they had a proficiency level of at least B1 in Italian. Moreover, they were administered LexIta [57], a lexical task assessing vocabulary competence through lexical decision. Participants who obtained a score over an established threshold (score ≥ 26, average B1 proficient participants' score at LexIta [57]) were selected for the iconicity rating task.

Table 1 provides information for age and gender for the two groups of participants.

To filter out unreliable raters, we applied a leave-one-out (LOO) reliability check. For each participant, we correlated their ratings on each word with the mean ratings of all other participants who had rated that word. This provides a measure of how well each participant's judgments align with the group, without circularity (since the target participant is not included in the mean). Only participants with a positive and significant correlation with the remaining participants (i.e., $r > 0$, $p < .05$) were retained. Out of 59 L1 participants, 54 passed this criterion (91.53%); out of 69 L2 participants, 57 passed (82.61%).

## Procedure

The study was implemented on Qualtrics and distributed online through Prolific or via e-mail to personal contacts and via snowball sampling. Participants were asked to rate a list of words (224 + 10 items each) in terms of how much they

**Table 1. Distribution of age and gender of the sample of participants that passed our filtering criteria (and thus were included in the analyses), divided by L1 and L2.**

|  | L1 | L2 |
|---|---|---|
| **Age (mean/std/min/max)** | 34.12/ 10.17/ 22/ 61 | 41.32/ 17.45/ 20/ 74 |
| **Gender (m/f/n.a.)** | 25/ 26/ 3 | 15/ 38/ 4 |

considered each word to be iconic. Instructions were presented in Italian and in English and were based on those used in the collection of iconicity ratings for English words [6]. The full text of instructions (Italian and English versions), along with analysis scripts and data, is available in the public online repository (see *Data Availability Statement* section). In the task, participants rated the words on a 7-point Likert scale and, for each word, they had the option to select a "I do not know this word" button and skip the rating judgment. (The extent to which L2 speakers pressed this button was negatively correlated with their proficiency; $r = -0.31$, $p = 0.009$).

## Analyses

**Reliability and first-level analyses.** As a first step, we checked whether the data we collected from both L1 and L2 participants was reliable and whether there were differences in reliability in the two data subsets. To do so, we split the available observations for each word into two random subsets and calculated the split-half correlation between the averaged word-by-word ratings in the two data subsets. For each dataset, this procedure was repeated 1,000 times with different random assignments of the observations to the two lists. Random seeds were set for reproducibility purposes. In addition to reporting the raw split-half correlation, we also indicate the results obtained by applying the Spearman-Brown correction.

**Iconicity ratings and psycholinguistic variables (L1 speakers).** Iconicity ratings have been shown to correlate with a range of psycholinguistic variables [6]. In our analyses, we examined how the iconicity ratings in Italian (as rated by L1 speakers) were related to the different variables that have been previously shown to influence the perceived iconicity of a word. We restricted our analyses to the L1 data since the various psycholinguistic variables we consider have been rated by L1 speakers.

In these analyses, we considered several lexical variables as measured by previous ratings studies, namely perceptual modality norms (visual, auditory, tactile, gustatory, and olfactory perceptual strength [56]), concreteness [30], and age of acquisition ratings [58]. Beyond these human-rated lexical variables, we also considered measurements based on the distribution of words and characters occurring in corpora of natural language use. In particular, we considered word log-frequency (derived from subtitle-based corpora [59]) and letter log-frequency (derived from the same source). We further considered another predictor, semantic density, which indicates the number of semantic neighbors of a given word. We calculated semantic density using FastText embeddings [60], high-dimensional representations of word meaning derived from word co-occurrence patterns in natural language (see [61] for an overview on the use of vector-space models in cognitive psychology). In line with Buchanan et al. [62], we operationalized the semantic density $D$ of a word $w$ as the average cosine similarity between $w$ and its 10 closest neighbors. More formally, we calculated the semantic density $D$ relative to $w$ as:

$$D = \frac{1}{10} \Sigma_{i=1}^{10} \frac{w \cdot n_i}{|w| \cdot |n_i|}$$

Where $n_i$ (for $i = 1, 2, \ldots, 10$) is each one of the closest semantic neighbors with respect to $w$. Additional details on the metric can be found in [61,62].

In addition to the variables that have already been shown to relate to iconicity in the previous literature, we also considered *specificity* [53], i.e., the degree of inclusiveness of the category of reference (e.g., highly generic: *animal* vs. highly specific: *okapi*). Specificity has been shown to explain variance in word processing times and has been proposed to be a core principle in the organization of semantic knowledge.

The relationship between all the variables reported above and perceived iconicity was assessed by means of a linear mixed effects model; the various predictors were all entered simultaneously in the statistical model aimed at predicting each L1 participant's rating. To account for the hierarchical structure of the data, the model was fitted with random intercepts for both participants and words. The model was fitted on a total of 13,910 responses, relative to 1,104 words, as the

remaining ones did not have ratings for some of the considered lexical features or were not present in the corpora from which we obtained log-frequency estimates for letters and words.

Additionally, we assessed whether the iconicity ratings we collected from L1 participants were predictive of word processing latencies and accuracy, using existing resources available for the Italian language. In particular, we relied on pre-existing data collected with a lexical decision task (LDT) and naming task (NT) for the words in the ANEW vocabulary [56], and lexical decision data collected in the context of a larger online crowdsourcing initiative [63]. The responses were analyzed through linear regression, predicting the dependent variable of interest (NT/LDT processing times or accuracy) with each word's average iconicity ratings. These models further included as linear covariates the various psycholinguistic variables described in the previous paragraphs, namely perceptual modality norms, concreteness, age of acquisition, letter (log) frequency, word (log) frequency, semantic density, and specificity. Since the focus of the analysis is the evaluation of the effects of iconicity on lexical processing, we will refrain from discussing the impact of the other covariates in the following sections.

**Relationship with the English ratings.** After evaluating the link between the L1 ratings and the various psycholinguistic variables described in the previous section, we assessed the correlations between the ratings produced by the L1 Italian speakers and the English iconicity ratings [6]. The Italian and English words were aligned based on the English translation of the Italian ANEW materials [30].

**Relationship between L1 and L2 ratings and effects of L2 proficiency.** The relationship between the ratings produced by L1 and L2 speakers was measured by means of Pearson correlation. For the L2 data, we further evaluated how their proficiency in Italian modulated the relationship between their ratings and the L1 and the English ones. Operationally, we examined the correlation between (a) each L2 participant's proficiency level (as assessed with the LexIta questionnaire, see "Participants") and (b) the correlation between that participant's ratings and the L1 and English ratings.

**Cross-linguistic visualization and hypothesis generation.** To gain qualitative insights into the distribution of iconicity judgments across L1 and L2 speakers, we standardized word-level means separately for L1-Italian and L2-Italian speakers and plotted each word in the L1–L2 plane. The plane was partitioned into four bins: high-L1/high-L2 rating (HH), low-L1/low-L2 (LL), high-L1/low-L2 (HL), and low-L1/high-L2 (LH). For emphasis, we computed an "extremeness" index for each word as the sum of absolute standardized coordinates and highlighted the ten most extreme items per quadrant. Qualitative inspection suggested two testable hypotheses: (i) some LH words appear phonologically similar to English translations (e.g., *miracolo–miracle*), predicting that greater cross-linguistic phonetic similarity should associate with higher L2 ratings; (ii) LH items also tended to be more frequent, predicting a positive association between frequency and L2 ratings.

**Phonetic distance and frequency.** We formalized these hypotheses with a residual-based partial correlation approach. First, we regressed the L2 word-level standard scores on the L1 scores and calculated the residuals; this residual captures variance in L2 not linearly predictable from L1—in other words, variance in L2 ratings that is specific to L2 speakers. We then correlated these residuals with (a) cross-linguistic phonetic distance and (b) word frequency.

**Phonetic distance.** We computed phonetic distance between each Italian word and its English translation using an IPA and feature-based pipeline: Epitran for grapheme-to-phoneme (IPA) conversion and PanPhon for weighted feature edit distance over segmental feature bundles. PanPhon [64] decomposes each IPA segment into a vector of articulatory features (covering major class, manner, place, voicing, and other properties). Word-level phonetic distance was calculated with PanPhon's weighted feature edit distance, which aligns the two IPA sequences and computes the minimum cost of transforming one into the other. This distance was normalized by the maximum string length of the two forms to make distances comparable across short and long words. Low values indicate high phonetic similarity; high values indicate phonetic divergence. For details, see [65] (Epitran) and [64] (PanPhon).

**Frequency.** We used the same frequency estimates used in the main analyses, derived from subtitle-based corpora [59]. We then computed Pearson correlations between the L2 residuals and (a) phonetic distance and (b) frequency. These correlations directly test whether, controlling for L1 ratings, L2 judgments are sensitive to cross-linguistic phonological similarity and frequency.

## Results

### Reliability and first-level analyses

The L1 data we collected displayed a moderate-to-high reliability, as estimated with the split-half correlation ($r = 0.5677$, SD $= 0.0711$); the Spearman-Brown corrected reliability (henceforth $r_{SB}$) was $r_{SB} = 0.7215$ (SD $= 0.0606$). The reliability was lower for the L2 data ($r = 0.4561$, SD $= 0.0807$; $r_{SB} = 0.6220$, SD $= 0.0806$).

### L1 Iconicity ratings and psycholinguistic variables

The coefficients associated with the various psycholinguistic variables described in "Iconicity ratings and psycholinguistic variables (L1 speakers)" are reported in Fig 1. The plot reports the standardized coefficients with their relative significance level. It shows that perceptual strength in the auditory ($\beta = 0.0832$, SE $= 0.0132$, $t = 6.3075$, $p < 0.001$) and haptic ($\beta = 0.0561$, SE $= 0.0181$, $t = 3.1089$, $p = 0.0019$) modalities is positively associated with perceived iconicity; among the two sensory modalities, auditory perceptual strength displays a stronger association with the dependent variable being analyzed. The same positive and significant association is found with specificity ($\beta = 0.0707$, SE $= 0.0171$, $t = 4.1335$, $p < 0.001$), indicating that words referring to more distinct and precise concepts were associated with a tighter relationship between sound and meaning.

Differently from the other perceptual variables, concreteness displayed a negative relationship with the construct under scrutiny ($\beta = -0.0562$, SE $= 0.0215$, $t = -2.6147$, $p = 0.009$); while this result might seem counter-intuitive at first, it is in line with what has been reported in English ( [6]; see the Discussion section for an in-depth argument). Words acquired early during development tended to be rated as more iconic ($\beta = -0.0348$, SE $= 0.0166$, $t = -2.1020$, $p = 0.0358$). Words that were more frequent ($\beta = -0.0616$, SE $= 0.0175$, $t = -3.5104$, $p = 0.0047$) and composed of frequent letters ($\beta = -0.0528$, SE $= 0.0119$, $t = -4.4163$, $p < 0.001$) were rated as having lower form-meaning resemblance. For what concerns the prediction of word processing latencies and accuracy, iconicity ratings were found not to be predictive of online processing measurements in any dataset and response measure we considered. In the dataset released as part of Vergallito et

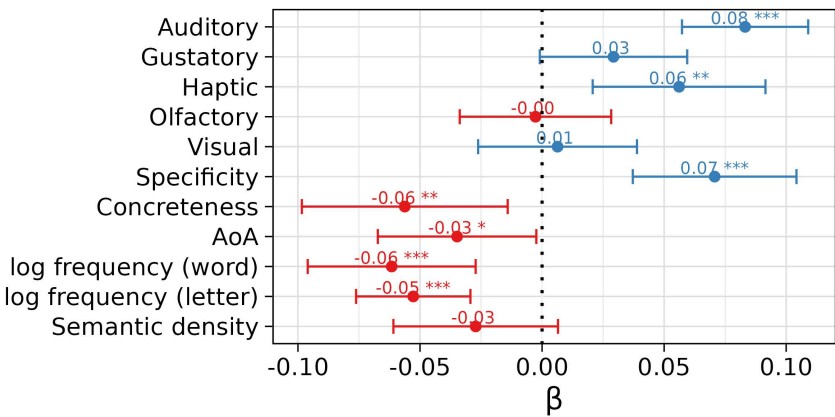

**Fig 1. Standardized coefficients for the various psycholinguistic variables predicting L1 iconicity ratings.** Associated significance level (* $p < 0.05$, ** $p < 0.01$, *** $p < 0.001$). The error bars indicate the coefficients' standard error.

al. [56], iconicity ratings were not associated with response times and accuracy scores neither in LDT (response times: $\beta=1.069$, SE = 2.741, $t=0.390$, $p=0.0696$; accuracy: $\beta=-0.003$, SE = 0.003, $t=-1.058$, $p=0.290$) nor NT (response times: $\beta=2.229$, SE = 2.328, $t=0.957$, $p=0.3386$; accuracy: $\beta=0.0002$, SE = 0.0003, $t=0.522$, $p=0.6018$). Similarly, when considering the psychometric resource provided by Amenta et al. [63], the ratings did not contribute significantly to the prediction of reaction times ($\beta=5.8624$, SE = 5.3312, $t=1.100$, $p=0.2718$) nor accuracy scores ($\beta=-0.0004$, SE = 0.0007, $t=-0.593$, $p=0.553$).

### Relationship between L1 and L2 Italian ratings and the English ratings

Both the L1 ($r=0.3124$, $p<0.001$) and the L2 data ($r=0.1884$, $p<0.001$) exhibited a significant and positive correlation with the English ratings [6].

### Relationship between L1 and L2 ratings and effects of L2 proficiency

The L1 and L2 data were positively correlated, as expected ($r=0.4292$, $p<0.001$). Critically, this correlation substantially exceeded the one observed between both the L1 and the L2 data and the English L1 ratings. Furthermore, the L2 participants' proficiency was found to be positively associated with the correlation between their ratings and the L1 ones ($r=0.34$, $p=0.001$) — in other words, more proficient L2 speakers produced ratings that resembled the L1 ratings. Proficiency was also positively associated with the correlation between the participants' ratings and the English ones, although this relationship was substantially weaker and not significant ($r=0.15$, $p=0.261$). A visual representation of these results is provided in Fig 2.

### Divergence between L1 and L2 ratings

To better understand where L1 and L2 judgments diverged, we constructed a quadrant visualization of the standardized mean ratings (Fig 3). Words were categorized according to whether they fell above or below the mean in both groups,

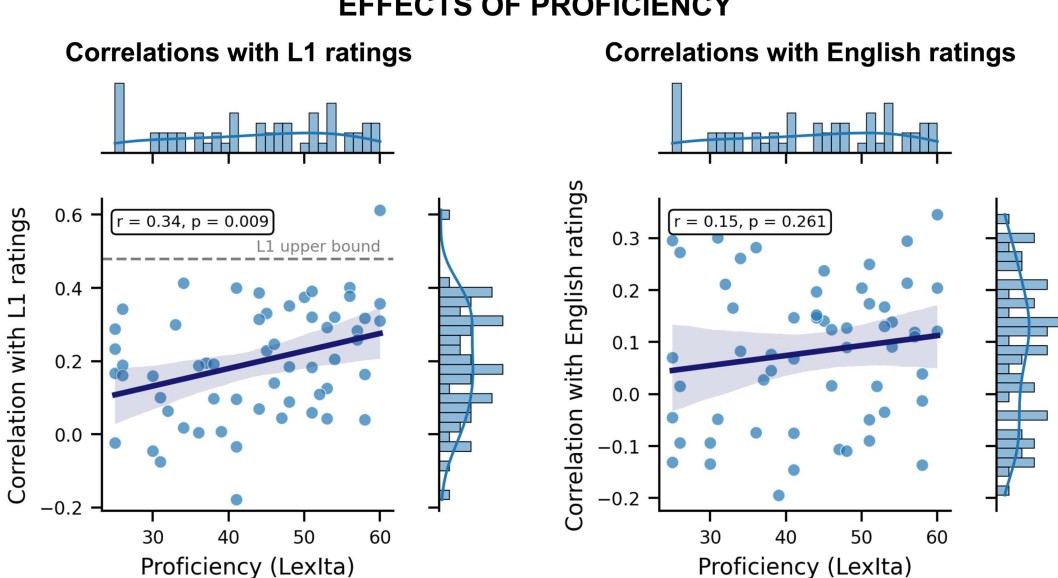

**Fig 2. Relationship between L2 proficiency and correlation with English and L1 ratings.** The *x*-axis represents the L2 proficiency score obtained from the LexIta test, while the *y*-axis indicates the correlation with L1 ratings (left panel) and with the English ratings (right panel). The dashed gray line in the plot on the left ("L1 upper bound") indicates the average correlation between the L1 participants and the averaged L1 ratings.

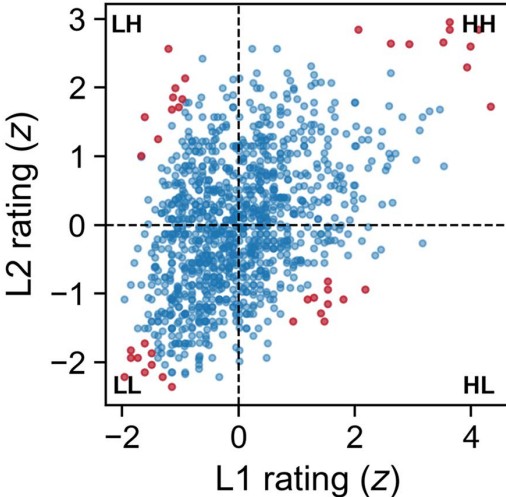

**Fig 3. Scatterplot of standardized L1 and L2 ratings.** Each dot is a word (blue = all words, red = top 10 most extreme words per quadrant). The dashed lines mark the zero point on each axis, dividing the space into HH, HL, LL, and LH quadrants.

yielding four categories: HH (high in L1 and L2), LL (low in both), HL (high in L1 but low in L2), and LH (low in L1 but high in L2). To highlight the most illustrative examples, we calculated an *extremeness* index as the product of the absolute standardized scores for each word, and we extracted the ten most extreme items in each quadrant (Table 2).

As expected, HH words (e.g., *bomba*, *stress*, *spray*) were obviously iconic items, while LL words (e.g., *mensola*, *mese*, *teglia*) do not display any apparent resemblance between form and meaning. More informative were the quadrants of divergence. HL words included metaphorical or figurative terms such as *abbaglio* (*dazzle*) and *tetro* (*dreary*), suggesting that semantic opacity may hinder L2 speakers' ability to perceive sound–meaning correspondence. LH words, by contrast, contained many items that closely resemble their English translations in form, such as *miracolo* (*miracle*), *idiota* (*idiot*), *elegante* (*elegant*), *unità* (*unity*), and *innocente* (*innocent*). Interestingly, these LH words also appeared to be higher in frequency than the HL items, pointing to two potential sources of L1–L2 divergence: phonological similarity to English and word frequency.

## Confirmatory analyses

The qualitative inspection of the cases of L1–L2 divergence suggested two hypotheses: (i) that L2 iconicity ratings are facilitated when Italian words are phonologically similar to their English translations, and (ii) that higher frequency words are judged as more iconic in L2. To test these hypotheses, we residualized L2 ratings against L1 ratings and correlated the residuals with word-level phonetic distance between Italian and English translations and lexical frequency.

The results confirmed both predictions. Residual L2 variability correlated negatively with phonetic distance ($r = -0.231$, $p < 0.001$), indicating that words more similar in sound across Italian and English received higher iconicity ratings from L2 speakers. In addition, residual L2 variability correlated positively with frequency ($r = 0.133$, $p < 0.001$), showing that more frequent words were also judged more iconic in L2.

## Discussion

In this work, we presented *IconicITA*, the first database of iconicity ratings for Italian. *IconicITA* provides iconicity norms for the 1,121 words of the Italian version of the ANEW database [30]. If the limited size of our database is surely a limit of the present work, this choice was motivated by the fact that it allowed us to correlate our iconicity ratings with various

**Table 2. Ten most extreme words in each quadrant of the L1–L2 space.**

| Category | Word | Translation | L1 Rating | L2 Rating |
|---|---|---|---|---|
| HH | caos | chaos | 5.25 | 5.33 |
| | stress | stress | 4.92 | 5.42 |
| | bebè | baby | 5.15 | 5.14 |
| | bomba | bomb | 4.92 | 5.33 |
| | spray | spray | 4.85 | 5.19 |
| | pizzicotto | pinch | 5.12 | 4.90 |
| | zanzara | mosquito | 4.46 | 5.17 |
| | charme | charm | 5.38 | 4.45 |
| | snob | snob | 4.25 | 5.18 |
| | orrore | horror | 3.88 | 5.33 |
| LL | mensola | mantel | 1.23 | 1.36 |
| | deludere | disappoint | 1.31 | 1.58 |
| | geranio | geranium | 1.46 | 1.42 |
| | mese | month | 1.31 | 1.67 |
| | prete | priest | 1.38 | 1.58 |
| | libro | book | 1.54 | 1.50 |
| | teglia | baking pan | 1.67 | 1.36 |
| | usanza | custom | 1.54 | 1.64 |
| | regalo | gift | 1.46 | 1.75 |
| | rivista | journal | 1.77 | 1.25 |
| HL | scolorina | ink remover | 3.50 | 2.00 |
| | arrugginito | rusty | 3.96 | 2.36 |
| | tetro | dreary | 3.71 | 2.25 |
| | disdegno | scorn | 3.46 | 2.09 |
| | abbaglio | dazzle | 3.54 | 2.20 |
| | re | king | 3.54 | 2.36 |
| | fangoso | muddy | 3.38 | 2.27 |
| | armadietto | locker | 3.14 | 2.00 |
| | colpire | hit | 3.31 | 2.25 |
| | spaventoso | fearful | 3.54 | 2.45 |
| LH | idiota | idiot | 1.73 | 5.11 |
| | miracolo | miracle | 1.46 | 4.33 |
| | eccellenza | excellence | 1.81 | 4.67 |
| | gusto | taste | 1.79 | 4.56 |
| | elegante | elegant | 1.92 | 4.78 |
| | unità | unit | 1.77 | 4.42 |
| | momento | moment | 1.88 | 4.54 |
| | scherzo | joke | 1.85 | 4.44 |
| | spaventato | scared | 1.62 | 4.08 |
| | innocente | innocent | 1.42 | 3.89 |

Values represent standardized ratings and the extremeness index.

psycholinguistic norms collected for the Italian version of the ANEW over the last decade [30,53,56,58]. To obtain our ratings, we employed a previously used methodology (e.g., [6,19,22]). However, instead of collecting ratings only from native speakers of Italian, we also collected ratings from speakers of Italian as L2, all sharing the same L1, English. This way, we were able to compare Italian L2 ratings not only with Italian L1 ratings, but also with English L1 ratings.

## Iconicity ratings for Italian by L1 speakers

Considering native speakers, iconicity ratings displayed a significant positive correlation with perceptual strength in the auditory modality. This replicates previous findings in English [22] and constitutes other evidence of a rather intuitive fact, i.e., that the degree of iconicity is maximal for words whose meanings strongly relate to the modality underpinning spoken language itself: audition. Considering spoken languages, concepts closely tied to auditory experiences naturally allow for greater resemblance between their acoustic form (phonetic structure) and their meaning. Moreover, L1 iconicity ratings correlated with perceptual strength in the tactile modality, further replicating English data [22]. This relation may seem less predictable than the one just described; however, several pieces of evidence suggest that, apart from audition, touch also has a close link with speech, also at the neural level ( [66], reported in [22]). Furthermore, a recent meta-analysis reported that touch-sound mapping is among the most dominant forms of linguistic synesthesia in the lexicon [67], reflecting systematic cross-modal integration and analogical correspondences between perceptual domains as central to the semantic elaboration of particular perceptual experiences (e.g., *smooth melody*). Such cross-modal and analogical correspondences also play a key role in structuring iconic mappings. Finally, one fact we can all reflect on, which directly links touch with hearing, is that touching something generally elicits a sound. This happens not only when we type on a computer keyboard or when we knock on a door, but also when we caress someone, and the list could go on and on. Even if tactile information is predominant in surface perception (e.g., [68]), touch-generated sounds can modify how a touch-based experience is reported (e.g., [69]), highlighting a link from the haptic to the auditory modality, which could be the key to understanding the positive relation between iconicity ratings and perceptual strength in the tactile modality. The third parallel result between Italian and English iconicity ratings is that there was no significant correlation between iconicity ratings and gustatory and visual perceptual strength. We also found no significant correlation between iconicity ratings and olfactory perceptual strength, whereas this relationship was negative and significant in [19]. In general, the lack of correlation we found between gustatory, visual, and olfactory perceptual strength can be seen as the flip side of the coin of what we discussed at the beginning of this paragraph, i.e., that the more a meaning relates to linguistic expression, the more the degree of iconicity is maximized. This reasoning further emphasizes the peculiarity of the tactile modality.

The correlation between iconicity and specificity was not included in previous analyses, as specificity ratings for English have only been released in late 2024 [70]. Nevertheless, previous descriptions of iconic words defined them as "linked to specific referents and contexts" [38]. Specificity indicates how precise a category is in defining its members, and what we found is that the more words are specific, the more iconic they are. Being iconicity a resemblance between form and meaning (or some aspect of), then the more a word activates a precise set of semantic (or perceptual) information, the more precise an iconic form-meaning mapping can be. As an example, consider two words from our dataset, *disturbare* (*to disturb*) and *zanzara* (*mosquito*). Both have a high auditory perceptual strength, but the former is a word with low levels of specificity and iconicity, whereas the latter displays high levels of both. It would surely be more difficult to find an iconic form to represent *disturbare* if we consider that both the noise of a mosquito and that of a pneumatic drill can be a source of disturbance, compared to *zanzara*.

L1 iconicity ratings were higher for less concrete words. Even if consistent with previous findings [6], this result remains puzzling. Why are less concrete words rated as more iconic? The concreteness dimension is particularly challenging because it tends to conflate different types of semantic information (e.g., perceptual, functional, and contextual), rather than capturing a single construct [52,71]. Indeed, Winter et al. [6], when discussing the negative correlation between concreteness and iconicity, consider *in primis* the validity of concreteness ratings. They claimed that such ratings might

be inclined towards visual perception, thus representing only a side of concreteness. Moreover, they discuss the fact that perceptual norms studies show a clear tendency for less concrete concepts to be perceived more through audition. The negative and significant correlation between concreteness and auditory perceptual strength found both in English [72] and Italian [56] might thus be the key to explaining this result.

Focusing on further negative correlations, Italian L1 speakers rated more frequent words as less iconic than less frequent words. In fact, it has been shown that, differently from children, adults use more non-iconic words than iconic words when speaking with other adults [11]. This negative relationship between iconicity ratings and frequency replicates findings based on English data [6]. Similarly, a negative correlation was found with log letter frequency, a measure of orthographic unexpectedness: L1 speakers rated words composed by more frequent letters as less iconic compared to words composed by less frequent letters, in line with previous research on English [6,73]. Letter frequency can be considered a measure of markedness, with words composed by frequent letters being unmarked, compared to words composed by less frequent letters [73]. The negative correlation between log letter frequency and iconicity indicates that iconic words tend to be marked as far as orthographic structure is concerned. Interestingly, this holds true even without considering onomatopoeias, as in our dataset.

Furthermore, we found a negative relationship between our iconicity ratings and AoA, revealing that earlier acquired words were, in general, rated as more iconic than words acquired later. This result is coherent with a wide range of research in several spoken and signed languages, highlighting the high prevalence of iconic words in the earlier acquired lexicon (e.g., [6,11,22,74]). A similar negative relationship, although not significant, was found between iconicity ratings and semantic density. Our data goes in the direction of previous findings on English [4,6], despite lacking significance. We suspect that this relationship might turn out significant also for Italian if a wider set of words is considered.

Finally, based on our L1 ratings, iconicity does not appear to be a strong predictor of RTs and accuracy in LDT. Considering spoken languages, only one work directly tested the effect of iconicity on word processing, showing a facilitatory effect of iconicity on decision latencies in LDT [43]. However, with the same set of 120 words, the authors did not find any effect with processing measures coming from the ELP [33]. Using instead a wider set of ELP data (the 2,852 words for which they had iconicity measures), the effect of iconicity was again significant. This makes our results less distant than those of [43] if the effect of iconicity is particularly affected by the specific data set considered. It is important to note, moreover, that the authors only considered as covariates word length, frequency, and orthographic neighborhood density (based on orthographic Levenshtein distance) and did not enter in the model all the covariates we used in the present study. Given that it is well-known that the perceptual content associated with a word influences its recognition times (e.g., [47,56,75,76]) and that iconicity is correlated with sensory experience ratings (particularly in the auditory modality), it is possible that sensory-related information might have driven the reported effect, as opposed to genuine sound-meaning resemblance. Another suggestion, that comes from studies on sign languages, is that effects of iconicity are to be expected when the iconic link between form and meaning is made salient by the task, thus making iconicity effects strategic effects, not linked to lexical access *per se* [77]. Understanding if this extends beyond the signed modalities requires carefully designed future studies.

## Iconicity ratings for Italian by L2 speakers

As we described in the Introduction, we decided to collect iconicity ratings from Italian L2 participants (English native speakers who learned Italian as a second language) with the goal of extending the debate on the construct validity of iconicity ratings. Specifically, we aimed to disentangle the relative contributions of word meaning and the form-meaning relationship in shaping participants' judgments by comparing Italian L2 ratings with 1) Italian L1 ratings, and 2) English L1 ratings. We hypothesized that, if iconicity ratings are primarily based on word meaning alone, correlations between Italian L2 ratings and both L1 groups should be comparable, as we focused on the same set of words translated from English into Italian. However, our results revealed that, although both correlations were significant and positive, the Italian L2 data

showed a stronger relationship with the Italian L1 data (moderate-to-strong correlation) compared to the English data (low-to-moderate correlation). This finding suggests that participants' judgments were influenced not only by the semantic properties of the words but also by language-specific phonological, morphological, or orthographic features. This seems to indicate that Italian L2 speakers accessed word meanings during the task, but their iconicity ratings were sensitive to the form-meaning relationship in the language of the task (i.e., Italian).

As a secondary objective, we explored whether higher proficiency levels enabled Italian L2 speakers to provide iconicity ratings more closely aligned with those of Italian L1 speakers. Previous research has emphasized the crucial role of language experience in recognizing iconic patterns [29]. This entails that proficiency should lead to a better appreciation of iconic patterns in the lexicon. Our results confirmed this hypothesis, showing that the more proficient a participant was in Italian, the stronger the correlation between their ratings and those of Italian L1 speakers. In contrast, when we examined the relationship between Italian proficiency and the correlation between participants' ratings and English ratings, we found a low and non-significant correlation. This latter result further suggests that, when providing iconicity ratings, participants relied primarily on form-meaning resemblance rather than on word meanings, and that their judgments were modulated by their proficiency in the language of the task, thereby supporting the validity of iconicity ratings.

Beyond the overall correlation between L1 and L2 ratings and the modulation by proficiency, our analyses revealed that the residual variance in L2 judgments is not random but systematically linked to theoretically meaningful properties. Words that were phonologically closer to their English equivalents, as well as words that were more frequent in Italian, received higher iconicity ratings from L2 speakers than predicted by L1 norms. This suggests that iconicity judgments in a second language are shaped by factors that facilitate semantic access. When an Italian word closely resembles its English translation, learners can recruit their knowledge of their native language to support meaning retrieval; when a word is frequent, repeated exposure strengthens its form-meaning association in memory. In both cases, easier access to meaning appears to increase the perceived naturalness of the link between form and meaning. These findings highlight that iconicity in L2 is not simply a direct reflection of L1 norms but is filtered through the mechanisms that govern lexical processing in a second language.

Finally, the fact that L2 participants were sensitive to language-specific mappings is in line with previous research on word learning in both L1 and L2 acquisition. In the case of children, more accessible form-meaning associations appear easier to learn. As we described in the Introduction, iconic words are among the earliest to be acquired and are frequent in child-directed speech (e.g., [10,38]). Similar effects can be found in adult L2 learners, where sensitivity to non-arbitrary sound-meaning relationships promotes the learning of novel word-meaning associations [78]. Taken together, these findings suggest that iconicity contributes to word learning, with potential implications for education. Highly iconic words may serve as useful entry points for learners: consider, for instance, that onomatopoeic forms are acquired very early, allowing lexical acquisition despite a still developing phonological capacity [79]. Iconic words can accelerate initial vocabulary growth and may further support the integration of less iconic or even non-iconic items, extending their facilitatory effect beyond early learning [80]. However, while this possibility of a global effect of iconic form-meaning mappings on language development is intriguing, further evidence is needed to verify this hypothesis [5].

## Conclusion

*IconicITA* is the first database of iconicity norms for the Italian language. This database contributes to further enhancing the list of psycholinguistic norms available for the 1,121 words of the Italian version of the ANEW [30], thus allowing to integrate iconicity into psycholinguistic investigation in Italian.

Regarding the relationship between iconicity norms and various psycholinguistic variables, we replicated many of the findings previously reported in the literature. When replication failed, the results were not drastically different but instead showed patterns consistent with prior literature, even though lacking statistical significance.

Compared to previous work, *IconicITA* presents the novelty of ratings collected also by L2 speakers. The correlation between L1 and L2 Italian ratings represents to us evidence towards the claim that iconicity ratings really do measure iconicity, even in a second language.

## Author contributions

**Conceptualization:** Andrea Gregor de Varda, Tommaso Lamarra, Andrea Amelio Ravelli, Chiara Saponaro, Beatrice Giustolisi, Marianna Bolognesi.

**Data curation:** Andrea Amelio Ravelli, Marianna Bolognesi.

**Formal analysis:** Andrea Gregor de Varda.

**Funding acquisition:** Marianna Bolognesi.

**Supervision:** Beatrice Giustolisi, Marianna Bolognesi.

**Writing – original draft:** Andrea Gregor de Varda, Tommaso Lamarra, Andrea Amelio Ravelli, Chiara Saponaro, Beatrice Giustolisi, Marianna Bolognesi.

**Writing – review & editing:** Andrea Gregor de Varda, Tommaso Lamarra, Andrea Amelio Ravelli, Chiara Saponaro, Beatrice Giustolisi, Marianna Bolognesi.

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
