## [Decision Letter · Decision Letter 0]

20 Jul 2025

Dear Dr. Lamarra,

Thank you for submitting your manuscript to PLOS ONE. After careful consideration, we feel that it has merit but does not fully meet PLOS ONE’s publication criteria as it currently stands. Therefore, we invite you to submit a revised version of the manuscript that addresses the points raised during the review process.

Thanks for your submission to PLOS One.  While this article describing iconicity ratings from L1 and L2 speakers for 1121 Italian words is timely and worthwhile, its theoretical and methodological bases require further elaboration. Specifically, research on iconicity ratings in ASL and by L2 learners should be described in the Introduction, and responses of "I do not know this word" should be analyzed, as should differences in exclusions of L1 and L2 speakers. Moreover, the impact of L2 proficiency on iconicity ratings requires further consideration, as does cross-linguistic comparisons of iconicity ratings between English and Italian. I encourage the authors to address these points as well as others raised by the reviewers in a revision, which I will attempt to send to the same reviewers to evaluate whether these points are addressed satisfactorily.

We look forward to receiving your revised manuscript.

Kind regards,

Laura Morett

Academic Editor

PLOS ONE

Journal Requirements:

2. Please note that your Data Availability Statement is currently missing the repository name and/or the DOI/accession number of each dataset OR a direct link to access each database. If your manuscript is accepted for publication, you will be asked to provide these details on a very short timeline. We therefore suggest that you provide this information now, though we will not hold up the peer review process if you are unable.

Reviewers' comments:

Reviewer's Responses to Questions

**Comments to the Author**

1. Is the manuscript technically sound, and do the data support the conclusions?

Reviewer #1: Partly

Reviewer #2: Yes

2. Has the statistical analysis been performed appropriately and rigorously?

Reviewer #1: No

Reviewer #2: Yes

3. Have the authors made all data underlying the findings in their manuscript fully available?

Reviewer #1: Yes

Reviewer #2: Yes

4. Is the manuscript presented in an intelligible fashion and written in standard English?

Reviewer #1: Yes

Reviewer #2: Yes

Reviewer #1: This ms presents iconicity ratings for 1121 Italian words, collected from two populations: L1 and L2 speakers. The number of words rated is quite low compared to other recent data sets (e.g. 13.000 for English in Winter et al. 2022), but the design does allow some interesting inferences and analyses, particularly because of the L1/L2 comparison. There is room for improvement particularly in the documentation of design decisions, scholarly embedding and in the description and interpretation of the L1/L2 comparison. I think the study requires revisions along the lines indicated below.

Methodology, documentation, analysis. It is good to have open data, but some aspects of the study are not yet documented in sufficient detail and others raise questions.

1. line 266 "Instructions were presented in Italian and in English". What were the instructions exactly? The supporting information that I could see included only an ethical approval form. We would need to see the full instructions as they were given, along with translations.

2. Line 268 "they had the option to select a "I do not know this word" button". The paper does not analyse these responses as far as I could see. Raters supplied on average 290 ratings (the mode is 234, some have double or quadruple that). Some have >90 "NA" values (presumably clicking "I do not know this word" a lot). There is sufficient variance among those NA rating numbers to wonder whether there is a relation to proficiency (presumably, less proficient speakers would more often not know some words). Matters like this need to be discussed and analysed.

3. Exclusion criteria and their effects. In an effort to filter out bad raters, a comparison was done between a set of words presumed to be highly iconic versus the rest; people who did not show a large enough difference between the two were excluded. But the disproportionately skewed effect of applying this criterion should raise alarm bells: only 2 L1 speakers were excluded (3% of participants) but as many as 26 L2 speakers (31%) (line 261). This is a finding in and of itself, unless for some reason the assumption is that L2 speakers are more likely to be "bad raters" (but no such argument is made). What this shows is that a significant portion of L2 speakers of Italian (of what proficiency?) have trouble distinguishing sets of words that according to L1 speakers should be highly distinct in terms of iconicity. That speaks directly to the goal of the study so the exclusion here seems not a great idea. At the very least more analysis of this very skew is required.

Questions about proficiency. Proficiency does not figure in the abstract and is not prominent in the theoretical background; it first appears as a subquestion in RQ2. But proficiency *is* the subject of one of the only 2 figures. More critical discussion of this is needed throughout. What is Figure 2 supposed to show exactly, and why? Would one assume that correlation would have the same direction in both cases? Essentially this suggests that the higher a person's proficiency in Italian, the higher there iconicity ratings of Italian words correlate with L1 English speakers' ratings of English words — why would that be the case? I found this confusing and it was not addressed in the text or caption relating to the figure.

Scholarly embedding I. Iconicity ratings were first introduced for signed languages, in particular American Sign Language (Griffith et al. 1981) and British Sign Language (Vinson et al. 2008). The earliest work by Griffiths et al. is particularly important to engage with because it also collected ratings for L1 and L2 users of the language, and so is directly comparable to the current ms. Much of this literature is reviewed in Ortega 2017.

Scholarly embedding II. Though as the study notes L1/L2 comparisons are relatively rare, this makes the few prior studies also looking into L2 iconicity ratings this all the more important to engage with. I am aware of two recent ones for a range of spoken languages. McLean et al. 2022 (uncited in the ms) collect ratings for Japanese words from English L2 learners of Japanese and compare them to the Japanese L1 ratings collected by Thompson et al. 2020 (cited). And Punselie et al. (2024) collect L2 iconicity ratings in 5 languages and report strong correlations between (a) L2 iconicity ratings, (b) experimental guessability, and (c) a direct measure of cumulative form-meaning associations. These lines of work supply complementary evidence that also helps to test and establish the construct validity of iconicity so it would be a good idea to cover them in the theoretical background.

Questionable assumption. The ms proposes that "correlations between Italian L2 ratings and both Italian and English L1 ratings should be comparable, as the meaning of a word remains constant across languages" (line 204-5). No evidence is cited for that bold claim about the stability of meaning; and in fact, there is a lot of work showing that meanings can be warped by language-specific semantic spaces (e.g., Werner 1993; Majid et al. 2007; Blasi et al. 2023). On p. 9 it becomes clear this comparison relies on the translation-based alignment provided in the Italian ANEW materials. It would be good to clarify in abstract (line 42) and research questions (line 204ff) and to choose a more cautious wording that avoids the questionable assumption that meanings are "constant across languages".

Interpreting the L1/L2 comparison. The main emphasis in RQ and abstract is on the L2 Italian to L1 English comparison. As the ms says, "Critically, this correlation substantially exceeded the one observed between [etc.]" (line 381). But the Figure in this section focuses on proficiency. Only the grey line in panel 1 shows a relation to the baseline L1 ratings. Somehow the rhetorics of the ms made me expect more information and a deeper analysis of how exactly the ratings differed across these two. E.g., one might expect an analysis of points of divergence and convergence. Which items were most dissimilar? which were most similar? and how might this relate to semantic domains, sensory modality, linguistic form, word class or other possibly relevant features? (Punselie et al. 2024 do something like this in their study of L2 ratings in relation to a bunch of other features. They find that for words with meanings in the domain with sound, there are fewer discrepancies across iconicity measures.) This is the kind of analysis that is uniquely possible in a study like this, and the ms delivered less of it than expected.

Copy-editing note: the numerical referencing style means that formulations that foreground the reference do not really work well. E.g., line 66 "Notably, [1] clarified that arbitrariness, systematicity and iconicity all contribute to language development and communication." This works if the authors are spelled out as in Chicago style, but for a numerical style it is better to move the reference to the end of the statement and lead with the finding: "Arbitrariness, systematicity and iconicity (...) [1]."

Nitpicking: line 121 "iconic words are the earliest words learned" — this is factually wrong. What the cited work and other work on AoA shows is that iconic words are relatively overrepresented among early words, meaning there are more of them than you might naively expect if words were encountered, used, and learned perfectly randomly. But there are still plenty of early words that are not all that iconic.

References cited

Blasi, Damián E., Joseph Henrich, Evangelia Adamou, David Kemmerer, and Asifa Majid. 2022. ‘Over-Reliance on English Hinders Cognitive Science’. Trends in Cognitive Sciences, October. doi:10.1016/j.tics.2022.09.015.

Griffith, Penny L., Jacques H. Robinson, and John M. Panagos. 1981. ‘Perception of Iconicity in American Sign Language by Hearing and Deaf Subjects’. Journal of Speech and Hearing Disorders 46 (4): 388–97. doi:10.1044/jshd.4604.388.

Majid, Asifa, Marianne Gullberg, Miriam van Staden, and Melissa Bowerman. 2007. ‘How Similar Are Semantic Categories in Closely Related Languages? A Comparison of Cutting and Breaking in Four Germanic Languages’. Cognitive Linguistics 18 (2): 179–94. doi:10.1515/COG.2007.007.

McLean, Bonnie, Michael Dunn, and Mark Dingemanse. 2023. ‘Two Measures Are Better than One: Combining Iconicity Ratings and Guessing Experiments for a More Nuanced Picture of Iconicity in the Lexicon’. Language and Cognition 5 (4). Cambridge University Press: 719–39. doi:10.1017/langcog.2023.9.

Ortega, Gerardo. 2017. ‘Iconicity and Sign Lexical Acquisition: A Review’. Frontiers in Psychology 8. doi:10.3389/fpsyg.2017.01280.

Punselie, Stella, Bonnie McLean, and Mark Dingemanse. 2024. ‘The Anatomy of Iconicity: Cumulative Structural Analogies Underlie Objective and Subjective Measures of Iconicity’. Open Mind 8 (September): 1191–1212. doi:10.1162/opmi_a_00162.

Thompson, Arthur Lewis, Kimi Akita, and Youngah Do. 2020. ‘Iconicity Ratings across the Japanese Lexicon: A Comparative Study with English’. Linguistics Vanguard 1 (ahead-of-print). De Gruyter Mouton. doi:10.1515/lingvan-2019-0088.

Vinson, David, Kearsy Cormier, Tanya Denmark, Adam Schembri, and Gabriella Vigliocco. 2008. ‘The British Sign Language (BSL) Norms for Age of Acquisition, Familiarity, and Iconicity’. Behavior Research Methods 40 (4): 1079–87. doi:10.3758/BRM.40.4.1079.

Werner, Oswald. 1993. ‘Short Take 10: Semantic Accent and Folk Definitions’. Cultural Anthropology Methods 5 (2): 6–7. doi:10.1177/1525822X9300500204.

Reviewer #2: In their manuscript, “IconicITA: Iconicity ratings of the Italian affective lexicon,” the authors quantify iconicity in Italian affective words using ratings made by L1 and L2 Italian speakers. Replicating findings in other languages such as English, they find that the ratings of L1 Italian speakers are positively associated with perceptual strength in the haptic and auditory domain and negatively associated with frequency and concreteness. As ratings data were previously not available in Italian, this manuscript offers and important resource to the field and the data are fully available to other researchers. Furthermore, the authors also address a novel and interesting question about whether L2 speakers’ ratings should correspond more with L1 speakers’ or with the ratings of words in their native language. Here they find that L2 ratings are more correlated with L1 ratings, a finding with implications for how people perceive and consider iconicity. Thus, I believe this manuscript will be of interest to the broad readership of PLoS ONE. I have a few suggestions aimed at improving the manuscript further.

The theoretical background section does a good job of citing the relevant extant literature. However, it would help the reader consider why the various patterns exist in English as described in this literature and are expected to exist in Italian, if the authors were to go into a bit more detail about the mechanisms at play here. E.g., why are auditory and haptic words more iconic, and why is there a negative relationship with concreteness if both concreteness and iconicity are associated with word learning, etc?

Can the authors comment on the connection between word learning literature in children learning an L1 and finding from L2 learners in the study. Are their implications for teaching a second language, e.g., targeting more highly iconic words?

**Do you want your identity to be public for this peer review?** For information about this choice, including consent withdrawal, please see our Privacy Policy

Reviewer #1: **Yes: ** M. Dingemanse

Reviewer #2: No

---

## [Author Response · Author response to Decision Letter 1]

13 Oct 2025

Please see the attached file named "R1_letter_response_to_reviewers" for our detailed responses to the reviewers' comments.

---

## [Decision Letter · Decision Letter 1]

16 Nov 2025

IconicITA: Iconicity ratings of the Italian affective lexicon

PONE-D-25-25201R1

Dear Dr. Lamarra,

We’re pleased to inform you that your manuscript has been judged scientifically suitable for publication and will be formally accepted for publication once it meets all outstanding technical requirements.

Kind regards,

Laura Morett

Academic Editor

PLOS ONE

Additional Editor Comments (optional):

I thank the authors for revising the manuscript in accordance with the reviewers' feedback. The manuscript is greatly improved from its previous instantiation and now meets the standards for publication in PLoS One. Therefore, I am pleased to recommend it for acceptance.

Reviewers' comments:

Reviewer's Responses to Questions

**Comments to the Author**

Reviewer #2: All comments have been addressed

2. Is the manuscript technically sound, and do the data support the conclusions?

Reviewer #2: Yes

3. Has the statistical analysis been performed appropriately and rigorously?

Reviewer #2: Yes

4. Have the authors made all data underlying the findings in their manuscript fully available?

Reviewer #2: Yes

5. Is the manuscript presented in an intelligible fashion and written in standard English?

Reviewer #2: Yes

Reviewer #2: Thank you for the opportunity to review this revised manuscript. The authors have done a good job responding to all of my previous questions and suggestions, and I find the manuscript much improved. I have no further suggestions.

**Do you want your identity to be public for this peer review?** For information about this choice, including consent withdrawal, please see our Privacy Policy

Reviewer #2: No

---

## [Editor Report · Acceptance letter]

PONE-D-25-25201R1

PLOS ONE

Dear Dr. Lamarra,

I'm pleased to inform you that your manuscript has been deemed suitable for publication in PLOS ONE. Congratulations! Your manuscript is now being handed over to our production team.

Kind regards,

on behalf of

Dr. Laura Morett

Academic Editor

PLOS ONE